# Evaluation of an Online Training Program on COVID-19 for Health Workers in Papua New Guinea

**DOI:** 10.3390/tropicalmed8060327

**Published:** 2023-06-19

**Authors:** Yasmin Mohamed, Priscah Hezeri, Hinabokiole Kama, Kate Mills, Shelley Walker, Norah Hau’ofa, Carmellina Amol, Madi Jones, Philipp du Cros, Yi Dan Lin

**Affiliations:** 1Burnet Institute, 85 Commercial Road, Melbourne, VIC 3004, Australiadani.lin@burnet.edu.au (Y.D.L.); 2Murdoch Children’s Research Institute, Flemington Road, Parkville, VIC 3052, Australia; 3Faculty of Medicine, Dentistry and Health Sciences, University of Melbourne, Parkville, VIC 3010, Australia; 4Burnet Institute, Kokopo P.O. Box 1458, Papua New Guinea; 5Johnstaff International Development, Lae 411, Papua New Guinea; 6School of Public Health and Preventative Medicine, Monash University, Melbourne, VIC 3004, Australia; 7National Drug Research Institute, Curtin University, Perth, WA 6102, Australia; 8Johnstaff International Development, Level 26 150 Lonsdale Street, Melbourne, VIC 3000, Australia

**Keywords:** virtual training, online training, training evaluation, COVID-19, Papua New Guinea

## Abstract

Background: Health worker training is an important component of a holistic outbreak response, and travel restrictions resulting from the COVID-19 pandemic have highlighted the potential of virtual training. Evaluation of training activities is essential for understanding the effectiveness of a training program on knowledge and clinical practice. We conducted an evaluation of the online COVID-19 Healthcare E-Learning Platform (CoHELP) in Papua New Guinea (PNG) to assess its effectiveness, measure engagement and completion rates, and determine barriers and enablers to implementation, in order to inform policy and practice for future training in resource-limited settings. Methods: The evaluation team conducted a mixed methods evaluation consisting of pre- and post-knowledge quizzes; quantification of engagement with the online platform; post-training surveys; qualitative interviews with training participants, non-participants, and key informants; and audits of six health facilities. Results: A total of 364 participants from PNG signed up to participate in the CoHELP online training platform, with 41% (147/360) completing at least one module. Of the 24 participants who completed the post-training survey, 92% (22/24) would recommend the program to others and 79% (19/24) had used the knowledge or skills gained through CoHELP in their clinical practice. Qualitative interviews found that a lack of time and infrastructural challenges were common barriers to accessing online training, and participants appreciated the flexibility of online, self-paced learning. Conclusions: Initially high registration numbers did not translate to ongoing engagement with the CoHELP online platform, particularly for completion of evaluation activities. Overall, the CoHELP program received positive feedback from participants involved in the evaluation, highlighting the potential for further online training courses in PNG.

## 1. Introduction

In Papua New Guinea (PNG), 44,811 COVID-19 cases and 663 COVID-19 deaths have been reported up to 10 August 2022, with underdiagnosis of community cases suspected [1,2]. Given the disproportionate burden of COVID-19 among laboratory staff and health care providers in PNG, particularly in 2020, it is crucial that health workers are supported to manage the outbreak of COVID-19 [3].

Health worker training is an essential component of a holistic outbreak response. While in-person training is preferred, national and international travel restrictions at the start of the COVID-19 pandemic in 2020 made face-to-face training particularly challenging. Virtual training is a useful alternative and has been shown to be successful in previous outbreak responses, including the Ebola outbreak of 2014–2015 in West Africa [4,5,6,7,8]. Several comprehensive COVID-19 training programs were implemented in PNG at the provincial and district level, led by the National Department of Health, the World Health Organization (WHO), and UNICEF (United Nations Children’s Fund) [3]. However, in mid-2020 there remained a need for targeted training aimed at a broad range of health provider cadres working in hospitals and health facilities in PNG.

As part of the PNG–Australia Partnership for Development, the Australian Government funded an online training program—the COVID-19 Healthcare E-Learning Platform (CoHELP)—implemented by the PNG National Department of Health, the WHO Representative Office for PNG, Johnstaff International Development (JID) and Burnet Institute. Technical input was also provided by Australasian medical associations and colleges (Table 1). Online training spread over the course of multiple months allowed for the development of shorter, more detailed modules, as well as frequent updates to training content in line with changing evidence from this novel disease.

Evaluation of training programs is essential for understanding their effectiveness on knowledge and clinical practice, as well as for informing the development of similar programs in the future [9,10]. We conducted an evaluation of CoHELP to assess the effectiveness of the online training program, measure engagement with the platform, and determine barriers and enablers to future implementation. The four levels of Kirkpatrick’s training evaluation framework underpinned the process of evaluation: [10,11] (1) reaction: to determine participants’ initial reactions to the training program and the overall acceptability; (2) learning: to determine what knowledge and skills were learned as a result of the training; (3) behaviour: to determine changes in clinical practice following the training; and (4) results: to determine how the training impacts on the participants’ broader area of work, including their team or department.

## 2. Methods

### 2.1. COVID-19 Healthcare E-Learning Platform (CoHELP)

The online CoHELP training program aimed to provide knowledge essential for frontline health workers and health management staff to respond to the COVID-19 outbreak in PNG. The program was delivered in two rounds in 2020 (round 1: 18 June–4 September; round 2: 18 September–18 December). Course components included an online training platform with 14 self-paced modules related to COVID-19, taking approximately 30 min per module (Table 1); one-hour lectures delivered once or twice weekly by health experts via video conferencing; additional downloadable resources (including local and international guidelines, checklists, and posters); access to recorded online training sessions; and an online closed discussion board to enable clinicians to ask questions of external experts. Participants could access materials within the course in any order after modules were released.

The evaluation team conducted a mixed methods evaluation (Table 2) using participant data from the online training platform, post-training surveys, qualitative interviews, and health facility audits. Aside from demographic data, all monitoring and evaluation tools were optional for training participants, to maximize the accessibility of the training materials. Examples of questions included in data collection tools are available in Appendix A.

Ethics approval was received from the PNG Medical Research Advisory Committee (MRAC No. 20.16) and the Alfred Hospital Ethics Committee (Project No. 478/20).

### 2.2. Online Training Platform Data

Data collected through the online platform included: a short demographic survey upon sign-up to the platform; a pre- and post-knowledge quiz at the beginning and end of each module to assess learning; and quantification of engagement with the online platform. Pre- and post-knowledge quizzes were a series of multiple choice and short answer questions that were asked at the start and end of the interactive online module to measure any changes in knowledge (assessing Kirkpatrick’s training evaluation level 2—learning). Descriptive analysis was performed and mean % change in pre- and post-knowledge quiz scores with standard deviation was calculated.

### 2.3. Post-Training Survey

An online survey was developed to assess opinions of the training program and opportunities for applying new knowledge and skills, using REDCap (Research Electronic Data Capture) tools hosted at Burnet Institute [12,13]. REDCap is a secure, web-based software platform designed to support data capture for research studies. The post-training survey was a series of multiple choice and short answer questions examining satisfaction with the training, changes in knowledge and clinical skills, and any perceived benefits and challenges of online training and applying training to practice. The survey assessed level 3 of Kirkpatrick’s training evaluation framework through self-reported changes in clinical practice. The survey link was emailed to all participants during the last month of the second round of training.

### 2.4. In-Depth Qualitative Interviews

In-depth interviews were conducted with fifteen participants of the CoHELP training, seven health workers who did not participate in the training (non-training participants), and six key informants involved in the project implementation. Interviews with training participants assessed level 1 of Kirkpatrick’s training evaluation framework by exploring acceptability of the training program.

For training and non-training participants, a sampling frame was developed to ensure a broad diversity of health worker roles, health facility type and geographical location. East New Britain, Eastern Highlands, Morobe, National Capital District and Western Province were purposively selected for logistical reasons and feasibility, representing areas where project staff were based and most CoHELP training participants were located.

Recruitment involved snowball sampling, whereby interviewees suggested other individuals whom they knew had participated in the training, and convenience sampling, which involved health care managers or those completing facility audits to provide names of known training participants for researchers to follow up with via phone call. It was our intention to interview six participants from each of the five provinces, however due to a range of logistical challenges, and resource and time constraints, in some areas this was not possible. The purpose of these interviews was to develop an in-depth understanding of experiences of participation in the training, including barriers to and enablers of participation.

Key informant interviews were conducted to gain an understanding of project implementation barriers and enablers. Key informants were purposively selected by the research team, to ensure a broad representation of experiences and perspectives, based on individual roles in the project. We approached eleven potential interviewees via email invitation; five were available for an interview; they included, included health professionals, academics, and donor organization representatives involved in project set up and implementation.

All interviews were conducted in English between December 2020 and February 2021 by a team of eight experienced researchers from PNG and Australia. The senior qualitative researcher (SW-PhD) conducted a half-day training session via Zoom to ensure consistent approaches were used to recruit participants and collect data. All researchers conducting interviews, except SW, were from PNG; SW only conducted interviews with key informants.

Interviews were between 15 and 45 min in duration. One researcher was present for each interview and took field notes as needed. Most interviews were conducted in-person at the health facilities; however, some training participant interviews occurred via telephone call and some key informant interviews were conducted via an online Zoom platform. Semi-structured interview schedules were used to guide all individual interviews. All interviewees were given gift vouchers to thank them for their time and contribution.

### 2.5. Facility Audits

Facility audits were designed to measure results of the training on level 4 of Kirkpatrick’s model (organizational change), concentrating on the application of infection prevention and control (IPC) principles in facilities, as these recommendations were most easily measured. Senior staff at the selected health facilities were contacted by project staff and invited to undertake an audit of their facility as part of the CoHELP evaluation. Those who agreed to participate were given hard copies of the facility audit to complete and instructions for returning the completed tools to the research team.

Facility audits were undertaken in five provinces and health facilities were purposively identified to include a range of referral, provincial and district hospitals.

### 2.6. Data Management and Analysis

All responses were voluntary and no identifying information was collected. Participants were allocated a unique identifier to allow for comparison across data collection methods. A statement of consent was included for all participants completing the online survey. Written informed consent was obtained for all facility audits and qualitative interviews.

Microsoft Excel version 16.36 (2020) and Stata 13 (StataCorp. 2013, College Station, TX, USA) were used for all quantitative data analyses.

Interviews were audio-recorded and transcribed verbatim. Qualitative data were organised and managed using NVivo Qualitative Data Analysis Software, version 12 (QSR International, 2020, Burlington, MA, USA). A thematic analysis was conducted by two researchers (SW, PH), which involved reading and re-reading transcripts to familiarise themselves with the data, and a process of categorisation that involved searching for patterns, consistencies, and discrepancies in the data set. Final themes were reached by consensus with the broader research team. All quotes in this article have been anonymised; any identifiable information has been removed.

## 3. Results 

A total of 364 participants from PNG signed up to participate in the CoHELP online training platform. Of the 339 participants who responded to the question, one quarter (85/339) were based in the National Capital District, 18% (61/339) in Morobe, 10% (35/339) in Eastern Highlands, and 9.7% (33/339) in Western Province. There was at least one participant from each of the 22 PNG provinces. Participant demographics are presented in Table 3.

### 3.1. Engagement with Online Learning Platform

A total of 40% (147/364) of participants completed at least one module, with more people completing Module 1 (39%; 141/364) versus Module 14 (2%; 8/364) (Table 4). The online platform captured the number of participants who watched the recorded seminars for each module, and during the second round of training, the project team documented the number of devices logged in to view the seminars live (Table 4). Almost two-thirds of participants who watched the recorded seminars were female (64%; 70/109) and 61% (66/109) had not previously completed another COVID-19 training.

Table 4 shows the number of participants who completed the pre- and post-knowledge quiz for each module, as well as the mean percentage change between the pre- and post-knowledge quizzes. For all modules, an increase in average scores was observed. Over half of training participants completed at least one pre- or post-knowledge quiz (57%; 209/364). A higher proportion of nurse/midwives (69%; 74/108) completed pre- and post-knowledge quizzes compared with medical staff (53%; 33/62). More participants who completed at least one quiz through the online platform were female (60%; 124/209) and had not undertaken previous COVID-19 training (57%; 120/209).

### 3.2. Post-Training Survey

Between 27 November and 18 December 2020, 24 training participants completed the online post-training survey. Of those who responded to the online survey, one third worked at a national referral hospital (8/24), 25% at a provincial public hospital (6/24), and 21% at a non-government organization (5/24). Most health facilities where the participants worked were in urban areas (75%; 18/24). More than one-third (38%; 9/24) of respondents were nurses or midwives, 17% were community health workers (4/24), and 13% worked in management or executive roles (3/24). The majority of respondents (63%; 15/24) had completed another training on COVID-19 in addition to CoHELP.

A summary of the results from the post-training survey is outlined in Table 5. The most common reasons for recommending the program to others included that the training was informative, useful, covered important topics, and helped to develop knowledge and skills. Respondents reported having used the knowledge and skills gained from CoHELP to train other health workers, provide health education to patients and in the community, and improve the infection control practices at their facility.

Reported changes to clinical practices were mostly related to maintaining infection prevention and control measures, adapting to new challenges presented by the COVID-19 pandemic, and rearranging health services to accommodate patients with suspected COVID-19. Organizational changes observed included implementation of new protocols, staff and resource allocations, changes in staff attitudes, and implementation of infection prevention and control measures. Enablers of changes to clinical practice and organizational change were similar—these included support from colleagues or supervisors, availability of resources and supportive policies. Similarly, barriers that impeded changes in individual clinical practice and organizational change included a lack of resources, high workload/inadequate staffing, and a lack of support from supervisors or colleagues. Another barrier to organizational change was lack of supportive policies.

### 3.3. Qualitative Data

Of the 22 training and non-training participants who completed in-depth interviews (Table 6), 16 were female and 6 were male. They included medical doctors, nurses, midwives, and other health professionals from a diverse range of health facilities (including provincial, district and referral hospitals), and geographical locations. More than half (n = 12) were nurses, including two midwives. Three key themes were identified via our analysis of qualitative interview data: (1) training relevance; (2) participation enablers; and (3) barriers to CoHELP participation. These themes highlight positive outcomes of participation in the training, factors that were important for encouraging participation and obstacles that made participation challenging.

### 3.4. Training Relevance

The usefulness and relevance of the CoHELP training to their work was recognised by all those interviewed who participated in the training.


*Actually, it has a lot of impact um in the workplace […] I am very conscious of washing my hands, cleaning my table every morning, before I put my things on the table, also to wear mask when I see that I’m with a lot of people.*
(Health Extension Officer, East New Britain)

Most participants said they had not received any other previous training about COVID-19 or the pandemic, and, of the few participants who had, all agreed the CoHELP training was much more comprehensive and thorough.

### 3.5. Participation Enablers

Enablers for participation in the training included the accessibility and extensive promotion of the training. Accessibility was amongst the most important aspects of the online training described by the interviewees, which was especially important for nurses and midwives who were working shifts and therefore could not attend seminars that were streamed live:


*For someone who is always busy—having the online training, its good. So, if I miss a webinar … I can still join later. For example, to go out for courses during your work time it’s quite difficult … but you still have the free time after hours to get through the replay.*
(Nurse, East New Britain)

Participants who enrolled in the training described finding out about it via various promotion mechanisms. Promotion of the CoHELP training on social media (Facebook) was a particularly effective mechanism.

### 3.6. Barriers to CoHELP Participation

Several barriers were described as either preventing individual enrolment or participation in the training, including busy workloads; limited or no access to Wi-Fi, mobile data, or technological devices; not having the technological knowledge; or simply not having known about the project.

A significant barrier for participation was time; for those who enrolled but did not complete all modules, most said this was the primary factor. Some participants explained how hospital wards were under-staffed and people were “too busy” caring for patients or they simply had too much work to do, and therefore did not have the time to attend live seminars.

Another significant barrier for participation identified by many participants (including key informants) was poor Wi-Fi connections and limited access to mobile data or smartphones. For some, this prevented them even enrolling in the training, and for others it meant once they were enrolled, they could not complete all modules:


*I was trying to get my colleague to do the training, but some of them have money constraint, they couldn’t buy their data to go in to do this online … the data thing was very expensive for some.*
(Nurse, Morobe)

### 3.7. Facility Audits

Six facilities participated in the facility audits—these were regional, district or provincial hospitals, with number of beds ranging from 109 to more than 900 per facility. In a third of hospitals (2/6), not all non-clinical staff had received training on infection prevention and control for COVID-19. Whilst most hospitals had implemented screening of all patients for symptoms of COVID-19, screening of visitors and staff was not uniformly implemented. Hand washing and cleaning of environmental surfaces was practiced in most hospitals; however, half of respondents were unsure of protocols relating to linen management and safe burials. Half of facilities (3/6) responded that personal protective equipment (PPE) was not continuously available for both clinical and non-clinical staff, and most facilities had a contingency plan that included reprocessing or extended use of PPE.

Whilst all facilities used a separate waiting area for testing of patients with suspected COVID-19, only two had separate treatment areas for confirmed COVID-19 patients, with only one hospital having single rooms available for isolation. Although half of facilities had mechanical ventilation, only one facility reported that the adequacy of ventilation had been checked and monitored. Physical distancing was enacted for patient beds in the wards, however one-third of facilities responded that physical distancing was not implemented in non-clinical areas, on ward rounds or during staff meetings.

## 4. Discussion

This is one of the first evaluations of online training on COVID-19 in PNG. We aimed to assess the effectiveness of the CoHELP training program, through evaluating engagement and knowledge gained and determining barriers and enablers to implementation. Key findings include that initially high registration numbers did not translate to ongoing engagement with the platform; lack of time and infrastructural challenges were common barriers to accessing the online training; there were documented examples of improvements in knowledge and self-reported changes to clinical practice; and participants appreciated the flexibility of online, self-paced learning.

The number of participants who signed up to the CoHELP platform over the six months of implementation was promising, with a diverse range of health worker roles, facility types and geographical locations. However, more than half of those who signed up failed to complete even one online learning module, and only a small proportion of participants completed pre- and post-knowledge quizzes. As expected, given the phased roll-out of the modules, more participants completed online components for earlier modules, with numbers tapering off for each subsequent module. Numbers of participants logged in to live seminars likely underestimates actual numbers, as many health workers gathered in groups at facilities to watch presentations together.

There are several potential reasons for low engagement with the online platform. Firstly, barriers such as limited access to the internet, devices and data likely hindered participation in the online platform and evaluation activities. Previous evaluations of online training courses for health workers in resource-constrained settings found similar accessibility challenges [14,15,16,17]. Feedback from the CoHELP evaluation suggests that addressing logistical issues, such as internet and data access, and lack of technical knowledge for using teleconferencing, would improve access and engagement. These findings are consistent with a recently published evaluation of another online training program in PNG for emergency department staff [17]. The CoHELP program did provide participants with access to free mobile data to support engagement; however, this may not have been widely known among those who signed up for the training.

A lack of time was also highlighted as a barrier to accessing the online platform. Having shorter and less data-rich content targeted at different disciplines may help to support engagement with the online platform. Advertising the various components of the training and how to access them may also be beneficial, as some participants were likely unaware that training sessions were recorded and could be viewed later. The broader challenge of a lack of time among training participants is considerably more difficult to address. As in many low- and middle-income settings, PNG faces a chronic shortage of health providers that has only been exacerbated by the COVID-19 pandemic [18,19].

For those completing modules and feedback, there was strong positive feedback overall for the course, with participants agreeing that it covered the priority topics, was relevant, and should be continued. Acknowledging the small sample size, this feedback was consistent across the multiple methods used in the evaluation, suggesting it is likely to be a true reflection. However, evaluation activities were not a compulsory part of the training program, and therefore those who chose to complete them may have had different opinions than those who did not.

For some training participants, the online modality was seen as an advantage of the training program, and the flexibility and self-paced nature of the training facilitated engagement with the platform. Improvements in quiz results following completion of online learning modules indicate knowledge gains, and responses from the post-training survey and the qualitative interviews suggest positive changes to clinical practice. It should be noted that the data collected mostly reflected self-reported improvements in knowledge and practice, with the inherent potential for response bias. The evaluation of the Emergency Care Systems Training Program in PNG also found improvements in knowledge and confidence among participants of a digital-based learning strategy [17].

Online survey results suggest the use of skills and changes to clinical practice; however, there was less evidence of change in organizational practice. This is expected, as the course was focused on individual learning with only a minority of participants indicating they held management roles. The highest level of Kirkpatrick’s training evaluation framework (impact on the broader area of work, such as team or department) is known to be particularly difficult to assess in the healthcare setting [20]. Although facility audits were designed to assess this, there are several limitations to using this tool as a measure of organizational change, in particular the engagement of staff from these facilities with the infection control module and the assumption from Kirkpatrick’s model that training results in behaviours that ultimately lead to positive organizational change, whereas facility audits show there are complex challenges (e.g., lack of personal protective equipment and lack of space) that impact ability to implement the application of infection control training. In developing future training programs, value could be sought by engaging with PNG clinicians and managers to determine what topics would be most useful, an approach that could maximize the applicability of training to resource-limited settings. However, access to international and best-practice guidelines can be used to advocate for health system improvements.

In order to prevent nosocomial transmission of COVID-19, including in non-clinical settings such as tearooms and offices, all health facility staff including non-clinicians need to be trained in infection prevention and control. Facility audit results showed that, while staff had been trained in the use of PPE in all facilities, not all staff had been trained in other principles of infection prevention and control. One quarter of participants chose “other” as their role in health care and 39% chose “other” as their department. From the qualitative interviews, these included non-clinical health workers such as data managers and chaplains. It is as important to train non-clinicians, as non-clinical staff have accounted for a significant proportion of workers infected with COVID-19 in hospitals [21,22]. Although the audits were only undertaken in six facilities in PNG, they provide a useful snapshot of the current state of infection prevention and control in hospital settings and highlight the need for additional training and support to this area.

This study had several limitations. Firstly, it was not possible for the evaluation team to observe behaviour change, clinical skills or demonstrate impact on organizational practices or patient outcomes. This is a common challenge with post-graduate medical training, including in-service training, and is not limited to online modalities [10]. Data collection was especially challenging with COVID-19 restrictions in place, potentially leading to selection bias at both the individual and the facility level. Most training participants were located in provinces where the collaborating organizations had strong professional networks, and therefore where the training platform was most promoted. Only a small proportion of training participants completed the evaluation activities. As these were optional, it is possible that those who did provide feedback viewed the program more positively than those who did not. Given the nature of data collection, social desirability bias may also have impacted the feedback provided by participants. A further limitation of this study is that there has not been a formal assessment of the reliability of the questionnaires. Finally, any changes to knowledge or self-reported clinical practice cannot be fully attributed to CoHELP, as additional training on COVID-19 was being undertaken across PNG in the same time frame. One important strength of this evaluation is the use of multiple methods of assessment, allowing triangulation of data and improving the validity of the findings.

## 5. Conclusions

Despite significant logistical and workload challenges, the positive feedback regarding the quality and relevance of CoHELP, and improvements in knowledge and skills following completion of the program indicate there is potential for online training to be a useful tool for meeting the needs of healthcare workers in resource-constrained settings, such as PNG. Online training can successfully link staff with experts all over the world. Challenges faced by online course participants, such as internet and data accessibility, should be considered when implementing future online training programs in PNG. Health workers involved in future online training programs should be encouraged and supported to remain engaged with the platform in order to maximize the benefits. With the continually evolving evidence on the COVID-19 response at the individual and health system levels, there is also an ongoing need for health worker training and capacity development. Our evaluation shows that online training can be a useful additional tool for supporting the national COVID-19 response in PNG and adds to the evidence towards the acceptability and effectiveness of digital-based learning strategies in resource-limited settings.

## Figures and Tables

**Table 1 tropicalmed-08-00327-t001:** Training modules and lead organization.

Topic	Expert Organization/s
Module 1: Introduction to COVID-19	Burnet Institute
Module 2: Infection Control Basics	Burnet Institute
Module 3: Principles of Outbreak Control	Burnet Institute
Module 4: Infection Control Management	Burnet Institute
Module 5: Clinical Management Basics	Burnet Institute
Module 6: Emergency Department	Australasian College for Emergency Medicine (ACEM)
Module 7: Pregnancy and Birthing	Royal Australian and New Zealand College of Obstetricians and Gynaecologists (RANZCOG)
Module 8: Adapting Essential Services	Australasian Society for HIV, Viral Hepatitis and Sexual Health Medicine (ASHM); Burnet Institute
Module 9: Clinical Management Advanced	Burnet Institute
Module 10: Critical Care	Australian and New Zealand Intensive Care Society (ANZICS)
Module 11: Child Health	Murdoch Children’s Research Institute
Module 12: Theatre Management	Australian College of Perioperative Nurses (ACORN)
Module 13: Diagnostics and Testing	PNG Institute of Medical Research
Module 14: Nursing	Burnet Institute

**Table 2 tropicalmed-08-00327-t002:** Overview of all data collection activities.

Timing	Data Collection Activity	Format	Target Group
During the training	Pre- and post-knowledge quiz	Multiple choice and short answer questions in online platform	All participants
	Engagement with online platform	% seminars watched; % quizzes completed; % online learning modules completed	All participants
After the training	Post training survey	Multiple choice and short answer questions emailed to participants in the last month of the training	All participants
	In-depth interviews and key informant interviews	Qualitative in-depth interviews in the last month of the training	Purposive sample of facilities and participants, a sample of health workers who did not take part in the training, and key stakeholders
	Facility audits	Facility-level checklists	Purposive sample of facilities

**Table 3 tropicalmed-08-00327-t003:** Participant demographics.

Variable	% (n/N)
Gender	
Female	55 (199/364)
Male	39 (141/364)
Role	
Nurse/midwife	32 (108/341)
Medical officer	18 (62/341)
Health extension officer	13 (43/341)
Community health worker	7 (24/341)
Management/administration	6 (19/341)
Other	25 (85/341)
Department	
Emergency	13 (44/341)
Hospital administration/management	15 (50/341)
ICU	4 (13/341)
Infectious diseases	11 (39/341)
Maternity/obstetrics	5 (18/341)
Surgical/operating theatre	6 (19/341)
Paediatrics	4 (12/341)
Services	4 (12/341)
Other *	39 (134/341)
Completed prior COVID-19 training	48 (175/364)
Completed prior online training	34 (123/364)

* Participants did not provide additional information on their department.

**Table 4 tropicalmed-08-00327-t004:** Engagement with online platform.

Module	Number of Training Participants (N = 364)				Mean % Change between Pre- and Post-Knowledge Quiz Score (SD)
	Completed Online Learning Packages	Viewed Seminars Live ^a^	Viewed Recorded Seminars ^b^	Completed Pre- and Post-Knowledge Quizzes	
Module 1: Introduction to COVID-19	141 (39%)	21 (6%)	95 (26%)	105 (29%)	10.8 (16.0)
Module 2: Infection Control Basics	93 (26%)	23 (6%)	52 (14%)	40 (11%)	5.0 (13.4) ^a^
Module 3: Principles of Outbreak Control	61 (17%)	32 (9%)	23 (6%)	59 (16%)	8.8 (11.4)
Module 4: Infection Control Management	49 (14%)	19 (5%)	25 (7%)	45 (12%)	6.9 (20.6)
Module 5: Clinical Management Basics	33 (9%)	27 (7%)	22 (6%)	32 (9%)	2.6 (7.6)
Module 6: Emergency Department	27 (7%)	26 (7%)	15 (4%)	25 (7%)	18.4 (25.8)
Module 7: Pregnancy and Birthing	24 (7%)	19 (5%)	9 (2%)	23 (6%)	32.2 (26.1)
Module 8: Adapting Essential Services	20 (6%)	21 (6%)	0 (0%)	19 (5%)	5.6 (25.8)
Module 9: Clinical Management Advanced	22 (6%)	20 (6%)	0 (0%)	19 (5%)	30.5 (23.4)
Module 10: Critical Care	12 (3%)	10 (3%)	11 (3%)	11 (3%)	9.1 (24.3)
Module 11: Child Health	12 (3%)	10 (3%)	0 (0%)	12 (3%)	26.7 (21.9)
Module 12: Theatre Management	8 (2%)	7 (2%)	5 (1%)	8 (2%)	5.0 (14.1)
Module 13: Diagnostics and Testing	14 (4%)	12 (3%)	5 (1%)	13 (4%)	3.8 (12.3)
Module 14: Nursing	8 (2%)	10 (3%)	3 (1%)	8 (2%)	31.8 (13.7)
At least one module	147 (40%)	-	109 (30%)	209 (57%)	-

^a^ Round 2 training only, ^b^ number of devices logged in to the seminar.

**Table 5 tropicalmed-08-00327-t005:** Results of post-training online survey.

Evaluation Component	Response	% of Respondents (n/N)
Satisfaction	The training was relevant to my current role	88 (21/24)
	The length of the course was appropriate	71 (17/24)
	The content of the course was appropriate	88 (21/24)
	I would recommend the training program to others	92 (22/24)
Application of knowledge and skills	I have shared information learnt from the training with others	83 (20/24)
	I have used the knowledge gained from the training in my current role	79 (19/24)
	I have used the skills gained from the training in my current role	79 (19/24)
	I have made changes to clinical practice as a result of the training	54 (13/24)
	I have observed organizational changes as a result of the training	63 (15/24)
Effectiveness	Since completing the training, I am prepared to manage a patient with COVID-19	100 (11/11)
	My organization is prepared for an outbreak of COVID-19	60 (12/20)
Logistics	The online platform was easy to use	58 (14/24)
	Challenges with accessing training online:	
	Network and internet	67 (16/24)
	Lack of sufficient data	54 (13/24)
	Difficulties navigating the platform	8 (2/24)
	Lack of scheduled or allocated time to complete the training	13 (3/24)
	Benefits of accessing the training online:	
	Flexible timing	79 (19/24)
	A choice of modules	58 (14/24)
	Availability of online resources	58 (14/24)
Usefulness of online training components	The online learning packages were very useful	90 (18/20)
	The seminars were very useful	72 (13/18)
	The pre- and post-knowledge quizzes were very useful	81 (13/16)
	The discussion boards were very useful	62 (8/13)
	The additional resources were very useful	56 (5/9)

**Table 6 tropicalmed-08-00327-t006:** Characteristics of participants and non-participants of individual interviews.

Location	No. of Participants	Participated in Training	Gender of Participants	Professional Role	Type of Facility
East New Britain	4	Yes (n = 3) No (n = 1)	Female (n = 4) Male (n = 0)	Nurse (n = 1) Midwife (n = 1) Research Officer (n = 1) Infection Control (n = 1)	District hospital (n = 2) Provincial hospital (n = 1) Community NGO (n = 1)
Eastern Highlands	2	Yes (n = 1) No (n = 1)	Male (n = 2) Female (n = 0)	Environmental Health Officer (n = 2)	Provincial hospital (n = 1) Community NGO (n = 1)
Morobe	5	Yes (n = 4) No (n = 1)	Male (n = 1) Female (n = 4)	Nurse (n = 4) Doctor (n = 1)	Provincial hospital (n = 5)
National Capital District	6	Yes (n = 5) No (n = 1)	Female (n = 6) Male (n = 0)	Nurse (n = 4) Midwife (n = 1) Chaplain (n = 1)	Referral hospital (n = 6)
Western Province	5	Yes (n = 2) No (n = 3)	Female (n = 2) Male (n = 3)	Nurse (n = 1) Community Health Worker (n = 1) Infectious Diseases (Data Officer/Counsellor (n = 3)	Provincial hospital (n = 4) Community NGO (n = 1)

## Data Availability

The datasets used and/or analysed during the current study are available from the corresponding author on reasonable request.

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
