# Peer review of "Evaluation of an Online Training Program on COVID-19 for Health Workers in Papua New Guinea"

_tropicalmed, 2023, doi:10.3390/tropicalmed8060327_

Round 1

Reviewer 1 Report

The authors have tried to assess a training program on multiple parameters. Likewise multiple tools are used to assess these parameters.

Topic is a mixture of multiple studies. Authors themselves have mentioned the evaluation as a mixed methods evaluation. The study does not provide any new evidence or fill any gap in the research field.

This study does not add anything new compared to previous published material. Despite being it a repetition of old studies, authors have jotted down the manuscript in a step wise manner, addressing one parameter at a time.

Results are valid to any training programe being conducted. Conclusions are consistent with the evidence that training module has benefit and positive feedback was received and same was the research question framed by authors.

Author Response

Thank you for the feedback. We believe that this study does provide new evidence. Firstly, there are few published evaluation studies of online training programs in resource limited settings performed during epidemics and pandemics, and this is one of the first for Papua New Guinea. Training evaluations often focus solely on pre and post assessments. We believe that the detailed methodology linked with a specific evaluation framework utilising multiple methods is also an important contribution to the literature. Finally, online learning is still a relatively new field, and while it has now been widely used in healthcare, evaluations that help to document challenges and benefits can be useful for future training conducted in similar settings. 

Reviewer 2 Report

Thank you for the opportunity to review this paper on an evaluation of an online training program in PNG. This paper is well written and has some important findings to inform future efforts to provide training in low resource settings. I suggest the journal accepts this paper after minor revisions. I have included some detailed comments below.

Introduction

-        In the aim, I suggest that you include ‘assess engagement/completion rates’ as this was something important that you did. I’m not sure about including ‘informing policy and practice responses to reduce the health impacts of the COVID-19 pandemic’ in your aim. This feels more like a potential outcome of the study, rather than a primary aim.

Methods

-        Can you provide some more information about how much time each module and the entire course took to complete? This is relevant for the discussion later on.

-        To help the reader, I suggest you put the items in Table 2 in the same order (and using the same terminology) as in the description in the methods

-        There is an imbalance between how much you describe the methods for the qual interviews versus the surveys – I suggest that you further describe the pre- and post-knowledge quiz to assess learning and the post-training survey

-        Please include your interview guide as supplementary materials

-        There is some repetition in the description of the qualitative interview study team that you could remove – see paragraph 1 and 4 in section 2.4

-        Can you provide some further information about your key informant interviewees – what roles did they play in the project?

-        Provide further information about the facility audit and include your questions as supplementary materials

-        Describe the consent survey participants gave

-        Describe what you did for the quantitative analysis

-        Section 2.5. – can you be clearer about the link between the results of the facility audit and the CoHELP training? Are you suggesting that the audit might indicate outcomes of the training? If you are, you will need to provide more detail, e.g. number of staff who received the training in these facilities, time elapsed since training and other relevant measures. If this is not what you are suggesting, can you be clearer about the purpose of this audit and what it tells us about the training. My take on this is that it might point to areas where future training could help, but again, I'm not sure how it relates to the current training. 

Results

-        You could improve the formatting of Table 3, the centre alignment makes it a little hard to follow, especially info in column 1

-        3.1 – it’s not clear to me why the denominator has changed from 364 to 360?

-        Table 4 – it would be useful to see the % and n/N – you already provide some of these proportions in the text.

-        For the knowledge quizzes, are you able to provide some information comparing how well different groups (e.g. midwives versus medical staff) did in the quizzes – this might tell us something about the training’s effectiveness or suitability for different groups.

-        To help the reader – use the same terminology throughout the paper when referring to your research tools e.g. write “pre- and post-knowledge quiz” rather than just “quiz”

-        For the qualitative interview results, provide the reader with a summary of themes (number, names) before describing the themes.

-        Were there any differences in categories and themes between participants in different geographical regions? What about between key informants and participants?

-        Re: the Facility audits, as per my comments in the ‘Methods’ section, I’m not sure what to make of these results, as it’s not clear what the aim of the facility audits was. Are you inferring that these results show the impact of the training? That is problematic as many of these issues have a lot to do with resource constraints and other issues not related to knowledge. Also, you mention that some of the staff did not receive training – be clearer about why you are mentioning this (as per previous comment, what was the aim of the audit?) and be clearer on whether all or some staff in the participating facilities received the CoHELP training.

Discussion

-        What was it about the provinces that had the majority of participants that enabled this larger number of participants? This could be something to discuss in terms of future training efforts

-        This sentence is problematic: “the training was effective amongst those who completed it and provided feedback” – I’m not sure you can make that conclusion as presumably you don’t have results from those who did not provide feedback?

-        This sentence – “There are several potential reasons for low engagement with the online platform. Firstly, excluding demographic details, all monitoring and evaluation data were optional for training participants to maximise the accessibility of the training materials” – this is a reason for low engagement with the evaluation, rather than with the training itself

-        You mention Kirkpatrick’s training evaluation framework in the methods, but don’t come back to it until you mention it briefly in the discussion. Can you provide some further information about how this framework informed your study design and analysis?

-        As per my comments earlier about the Facility audits, you write that “Results from the facility audits reiterate the need to engage with managers and administrators to address broader resource constraints and advocate for infrastructural and supply chain improvements.” – I agree this is an important issue, but it’s not clear to me how this is related to the training.

-        A lot of your findings seem relevant to providing training in healthcare settings at any time – can you discuss how your findings are relevant to providing training in emergency settings (like the pandemic) versus business as usual settings?

-        You mentioned ‘informing policy and practice responses’ in your aim, but it’s not clear how you achieved this aim – either in the results or the discussion. As I mentioned earlier, perhaps this is not a primary aim of your study, but rather a potential benefit of conducting a study such as this.

Conclusions

-        You write that “Online training can fit into the schedules of busy health workers and successfully link staff with experts all over the world” – but your findings suggest otherwise

Reviewer 3 Report

line 49: write the full sentense before the abbreviation

in method section: make flowchart demonstrate the steps of online training program with number of participants in each stage

what about the validity and reliability of the online survey questions also for indepth interview questions, what about pilot study/

line 159-160: what about the ethical commitee approval?,   only informed consent from the paricipant were mentioned

Table 3: demonstrate what other refer o as a foonotes below the table.

table 4: is there significan changes between pre and post training (p value??)
